# Small in size, big on taste: Metabolomics analysis of flavor compounds from Philippine garlic

Ralph John Emerson J. Molino[1][☯], Klidel Fae B. Rellin[1][☯], Ricky B. Nellas[2], Hiyas A. Junio[1]*

1 Secondary Metabolites Profiling Lab, Institute of Chemistry, College of Science, University of the Philippines, Diliman, Quezon City, Metro Manila, Philippines, 2 Virtual Biochemical Explorations Lab, Institute of Chemistry, College of Science, University of the Philippines, Diliman, Quezon City, Metro Manila, Philippines

☯ These authors contributed equally to this work.
* hajunio@up.edu.ph

**Data Availability Statement:** All relevant data are within the paper and its Supporting information files.

## Abstract

Philippine garlic (*Allium sativum L.*) is arguably known to pack flavor and aroma in smaller bulbs compared to imported varieties saturating the local market. In this study, ethanolic extracts of Philippine garlic cultivars were profiled using ultra-high performance liquid chromatography-quadrupole time-of-flight mass spectrometry (UHPLC-QTOF). γ-Glu dipeptides, oligosaccharides and lipids were determined in Philippine garlic cultivars through bioinformatics analysis in GNPS Molecular Networking Platform and fragmentation analysis. Multivariate statistical analysis using XCMS Online showed the abundance of γ-Glu allyl cysteine in Batanes-sourced garlic while γ-Glu propenyl cysteine, γ-Glu methyl cysteine, and alliin are enriched in the Ilocos cultivar. Principal component analysis showed that the γ-Glu dipeptides found in local garlic influenced their distinct separation across PC1 from imported varieties. This presence of high levels of γ-Glu dipeptides and probiotic oligosaccharides may potentially contribute to the superior flavor and nutritional benefits of Philippine garlic.

## Introduction

Garlic (*Allium sativum* L.) is a high-demand crop from the family Amaryllidaceae valued for its nutritional and therapeutic properties [1]. A culinary staple, this bulbous spice has been extensively studied [2] for its cardioprotective, anticancer, antidiabetic, and antimicrobial effects often attributed to the sulfonated compound allicin, which is produced from the crushing of the cloves [2]. In addition, previous reports have attributed garlic's aroma to allicin and its volatile degradation products [3,4]. Aside from these compounds, Ueda et al in 1990 isolated and identified γ-L-glutamyl-S-(2-propenyl)-L-cysteine (γ-L-glutamyl-S-allyl-L-cysteine), γ-L-glutam-yl-S-(2-propenyl)-L-cysteine sulfoxide, and glutathione [5]. Sensory evaluation of these compounds showed that these peptides exhibited flavor-modifying effects that are less

**Funding:** The authors received no specific funding from agencies in the public, commercial, or not-for-profit sectors for this work.

**Competing interests:** The authors have declared that no competing interests exist.

pronounced but comparable to glutathione [6]. Furthermore, these peptides, owing to their sulfur-containing structure, have intrinsic flavor that may be responsible for their flavor-modifying effect [6].

The Philippines is known to produce several garlic cultivars that are smaller in size but more pungent and aromatic than imports from China and neighboring Southeast Asian countries [7]. These cultivars are specifically found in the dry regions of the country such as in Batanes Province and Ilocos Region in the north, and the Mindoro Island in the south [8]. Native garlic cultivation is the main source of livelihood for farmers in this region [9]. Interestingly, a study in 2018 reports that locally produced garlic performs poorly compared to imported white garlic varieties more commonly found in the market [10]. This setback was attributed to the heavier and cheaper garlic produced from neighboring Asian countries that puts local garlic farmers at a disadvantage [11]. In fact, in 2020, garlic importation dominated the local production by 60,000 metric tons [12].

A 2019 study [13] differentiated four garlic cultivars grown in Mindoro according to their morphological attributes. Using size classification, Batanes white cultivar was noted to be the largest followed by Ilocos White, Lubang White, and Mindoro White. Also, the maturation of local garlic is faster; making its harvest 30 days earlier than its imported counterparts which are usually cultivated for at least 90 days. [14]. Moreover, there is currently limited knowledge on the chemical constituents of locally grown garlic across the Philippines.

Liquid chromatography-mass spectrometry (LC-MS)-based metabolomics presents a sensitive and high-throughput platform to study Philippine garlic cultivars [15]. Annotation of metabolites can be performed in the level of crude extract through Data-Dependent Acquisition (DDA) [16]. Spectral analysis can be carried out in bioinformatics platforms such as Global Natural Products Social Molecular Networking (GNPS) [17]. Furthermore, multivariate statistical analysis in the cloud-based XCMS Online [18] can be employed to explore similarities and differences between the native and the imported garlic cultivars in terms of their metabolite profiles. A detailed understanding of the phytochemistry of Philippine garlic could give insights on its sensory attributes and nutritional benefits. Furthermore, definitive comparison of native and imported garlic cultivars could provide the basis in promoting cultivation and trade of Philippine garlic.

## Materials and methods

### Sample preparation

Authenticated local cultivars were obtained from a farm in Ilocos Norte (ILAU samples; 18.1960˚ N, 120.5927˚ E), and from the Department of Agriculture Regional Office 2—Batanes Experimental Station in Basco, Batanes (BAU; 20.4634˚ N, 122.0042˚ E). These samples were authenticated based on seed or clove morphology. Native garlic was sourced from several locations: in Laoag City, Ilocos Norte (LA), and from the municipalities of San Jose (OMSJ; 12.3474˚ N, 121.0659˚ E) and Sablayan, Occidental Mindoro (OMSB; refer to S1 Figure 1 in S1 Fig). Meanwhile, garlic from Sablayan (12.8564˚ N, 120.9101˚ E), Occidental Mindoro, and in Zamboanga City (6.9214˚ N, 122.0790˚ E) in Mindanao were ascertained by retailers to be imported (IMPSB and ZAM respectively). Garlic samples of unverified origin were also bought in Sablayan, Occidental Mindoro (UNKSB), and in NEPA Q-Market, Quezon City (UNKQC; 14.6178˚ N, 121.0572˚ E; refer to S1 Figure 1 in S1 Fig) and included in this study. Samples were transported to the laboratory under ambient conditions.

Morphometric measurements involved the selection of five garlic bulbs per sampling location as biological replicates. The length, width, and thickness of each clove as well as the number of cloves per bulb were determined (S1 Table 1 in S1 Table). The authenticated Batanes

native cultivar has 12 or more smaller cloves, a trait shared with the native Ilocos variety. Most imported cultivars that have been naturalized in the Philippines have bigger but fewer cloves per bulb.

For each garlic cultivar from one location, five different bulbs were sampled which corresponds to the biological replicates. Fresh garlic cloves were crushed in absolute ethanol (USP grade, Merck® Germany, and Scharlab SI, Spain) at a ratio of 3:10 (g/mL). Setups were left to soak for three days at room temperature (28°C). Further profiling experiments revealed that prolonged soaking does not result in the degradation of metabolites (S2 Figure 1 in S2 Fig). Extracts were dried *in vacuo* and resuspended in LC-MS grade methanol (Merck® LiChrosolv®, Germany) and the same sample mass was resuspended to a final concentration of 1.0 mg/mL for analysis.

## LC-MS and MS$^2$ analysis

Positive ion mode LC-MS analysis of garlic extracts were analyzed using a Waters® Acquity UPLC® H-Class system hyphenated to a Xevo® G2-XS Quadrupole Time-of-Flight (qTOF) mass spectrometer with an electrospray ionization (ESI) source. Calibration of the qTOF was done prior to runs using mass calibrants and reference standards suitable for the LockSpray™ of Waters® MS.

Chromatography was performed through an Acquity UPLC® CSH Fluoro Phenyl column (1.7 μm, 50 mm long, 2.1 mm I.D.) maintained at 30°C. Acetonitrile (Merck® LiChrosolv® Hypergrade, Germany) and water (Merck® LiChrosolv®, Germany) infused with 0.1% formic acid (Pierce®, Invitrogen, USA) were used as binary mobile phase. The flowrate was kept at a constant rate of 0.350 mL/min with the following $H_2O$:$CH_3CN$ elution gradient: 95:5 at 0.75 min, 75:25 at 1.0 minute, 50:50 at 2.0 minutes, 20:80 at 2.25 mins, 0:100 at 4.50 mins, and then re-equilibrating back to 95:5 at 5.0 to 5.50 minutes. The injection volume for all samples is 1.0 μL. Each biological replicate is analyzed five times to provide technical replicate profiles for statistical analysis.

ESI was carried out in the positive ionization mode using the following settings: capillary voltage at 3.0 V, cone voltage at 42 V, and source offset of 80 V. Source temperature was maintained at 150.0°C, and desolvation gas temperature at 500°C. Full Scan (MS$^1$) analyses were done within the mass range of m/z 50.0 to 1500.0 and with a scan time of 0.50 second. These parameters were fine-tuned using an external standard within the mass range to improve sensitivity and ensure the accuracy and precision of the qTOF in mass detection.

MS$^2$ spectra were collected with the fast DDA mode of the instrument, acquiring MS$^1$ and MS$^2$ spectra at the range of m/z 50–1500, and a scan time of 0.50 second. MS$^2$ acquisition is initiated if ion intensity in the full scan exceeds the $3.0 \times 10^5$ threshold. Return to MS$^1$ scanning is triggered by the same total ion intensity of product ions, or after 0.25 seconds of MS$^2$ acquisition. A maximum of eight ions per scan were selected for MS$^2$ analyses. The precursor ions were subjected to collision-induced dissociation using argon curtain under fixed collision energies of 6.0 eV, 10.0 eV, and 15.0 eV as well as collision energy ramps of 15.0–30.0 eV, 30.0–45.0 eV, and 45.0–60.0 eV. Additional monitoring involving profiling of QC samples within two days of initial data acquisition was also performed.

## Multivariate statistical analysis

Multivariate analysis of MS$^1$ centroid data was done using XCMS Online [18]. Raw data was converted to 64-bit open-source mzxml via ProteoWizard MSConvert [19]. GUI Sets of multi-group analyses were performed for local authentic, local, and imported market samples as well as with garlic of unverified geographic origins.

Feature detection parameters include 5.0 ppm maximal tolerated m/z deviation, minimum and maximum peak width of 2.0 and 25.0 s minimum m/z difference of 0.01, signal-to-noise threshold of 10.0, prefilter intensity set to $1.0 \times 10^5$ and noise filter of 100.0. These parameters were based on the XCMS-suggested values for Waters high-resolution data [18] and were optimized for the samples in this study. The median-fold-change normalization is built in on the XCMS workflow for peak integration.

Kruskal-Wallis non-parametric analysis was selected as the statistical test with a p-value threshold for highly significant and significant features set to 0.01 and 0.02 respectively, with fold change for highly significant features greater than 1.5. Non-parametric analysis was performed as it does not rely on the distribution of metabolite features across the samples [20]. This is suitable for untargeted metabolomics data since the variances within the population are heteroscedastic [20].

Annotation parameters include 5.0 ppm error and 0.005 m/z absolute error. A width of 100.0 s is considered for extracted ion chromatograms and mass calibration gaps were also corrected. Non-metric multidimensional scaling (NMDS) and Principal Component Analysis (PCA) of MS$^1$ data sets were performed by XCMS Online as part of the multigroup analysis. NMDS and PCA scores and loadings plots were replotted using Xmgrace. Parameters for the statistical analysis were optimized according to the samples based on the initial suggested values of XCMS.

## Putative compound identification and molecular networking

MS$^2$ data were subjected to library matching in the cloud-based bioinformatics platform Global Natural Products Social Molecular Networking (GNPS) [17]. Criteria for library matching were set as follows: precursor ion mass tolerance of 0.45 Da, fragment ion mass tolerance of 0.50 Da, a minimum cosine (similarity) score of 0.70, and minimum matched peaks of 6. Same platform was used to create molecular networks that aided in the annotation of the metabolites. Parameters used for the generation of molecular networks include a minimum similarity (cosine) of 0.70 calculated from the fragmentation pattern of two precursor ions, six matched peaks, and a maximum number (top K) of 7 neighbors. Values for these analyses were optimized according to the type of instrument and data acquired for this study. Network visualization was customized using Cytoscape 3.7.1 [21].

## Results and discussions

### Metabolite profiling, putative compound identification and molecular networking

Untargeted metabolomics of authenticated native Philippine garlic (BAU and ILAU) reveals the presence of a broad range of compounds (Fig 1). Polar components ($t_R = 0.30$–$0.60$ min) that were found to be present were amino acids and oligosaccharides (e.g. sucrose, 1-kestose, and stachyose). Polar dipeptides ($t_R = 1.30$–$2.00$ min) were putatively characterized based on their fragmentation pattern, and comparison with the mass spectra of annotated γ-Glu dipeptides and available literature information [22,23]. Non-polar region ($t_R = 3.00$–$4.00$ min) represents the lipid peaks, consisting of fatty acids, phosphoethanolamine-, and phosphocholine-type lipids. A list of all metabolites putatively annotated by GNPS is presented in Table 1.

Majority of the compounds identified matched with gold spectra in the GNPS database. Library spectra categorized as 'gold' were derived from purified metabolites with nuclear magnetic resonance and crystallographic data [17]. Fragmentation patterns of GNPS-annotated metabolites can be referenced by precursor ions with no library database hit. Through

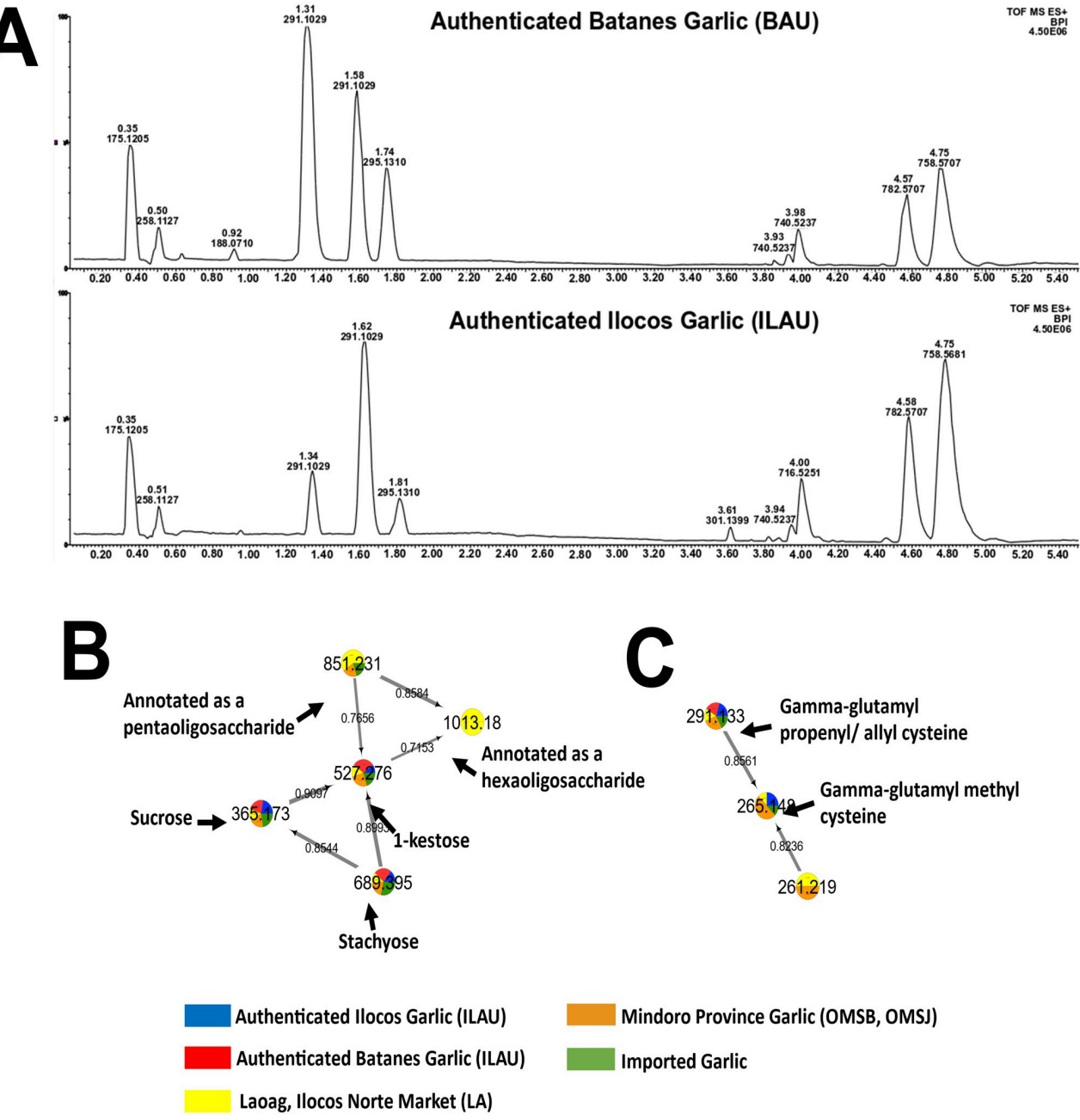

**Fig 1. Base peak chromatograms of authenticated garlic samples and resulting molecular networks with putatively annotated oligosaccharides and dipeptides.** Full-scan profiles of authenticated Batanes (BAU) and Ilocos (ILAU) garlic were shown in (A). GNPS generated networks with putatively annotated oligosaccharides and γ-Glu dipeptides are illustrated in (B) and (C), respectively.

fragment matching and molecular networking, structural relationships of two precursor ions can be scored for similarity with a cosine score of 0.00 to 0.99 representing no to very high similarity, respectively [17]. However, since the DDA mode only samples highly abundant metabolites, molecular networking such as shown in Fig 1B and 1C should be used with caution in deciphering the absence or presence of a metabolite in a group of samples. Nevertheless, these generated networks are useful in providing insights pertaining to the structural relationships

**Table 1. Secondary metabolites putatively annotated from authenticated BAU and ILAU garlic through GNPS analysis.**

| | Compound Name | $t_R$ (mins) | Major Ion | Experimental Mass | Monoisotopic Mass | ppm Error[a] | Cosine Score[b] |
|---|---|---|---|---|---|---|---|
| 1 | Arginine | 0.33 | [M+H]+ | 175.1205 | 175.1195 | 5.70 | 0.89 |
| 2 | 1-Kestose | 0.47 | [M+Na]+ | 527.1575 | 527.1588 | 2.48 | 0.92 |
| 3 | Stachyose | 0.48 | [M+Na]+ | 689.2119 | 689.2116 | 0.39 | 0.83 |
| 4 | Annotated pentaoligosaccharide | 0.48 | [M+Na]+ | 851.2628 | 851.2645 | 1.93 | MN/FA |
| 5 | Annotated hexaoligosaccharide | 0.48 | [M+Na]+ | 1013.3135 | 1013.3173 | 3.72 | MN/FA |
| 6 | Sn-glycero-3-phosphocholine | 0.5 | [M+H]+ | 258.1113 | 258.1106 | 2.52 | 0.85 |
| 7 | Melibiose | 0.52 | [M+Na]+ | 365.1064 | 365.1060 | 1.14 | 0.84 |
| 8 | Alliin | 0.60 | [M+H]+ | 178.0540 | 178.0538 | 1.12 | 0.74 |
| 9 | γ-Glu methyl cysteine | 0.92 | [M+H]+ | 265.0849 | 265.0858 | 0.91 | MN/FA |
| 10 | Indole-3-lactic acid | 1.07 | [M+H-H2O]+ | 188.0710 | 188.0711 | 0.81 | 0.88 |
| 11 | γ-Glu allyl cysteine | 1.33 | [M+H]+ | 291.1029 | 291.1015 | 4.92 | MN/FA |
| 12 | γ-Glu propenyl cysteine | 1.59 | [M+H]+ | 291.1029 | 291.1015 | 4.92 | MN/FA |
| 13 | γ-Glu phenylalanine | 1.74 | [M+H]+ | 295.1310 | 295.1294 | 5.43 | 0.88 |
| 14 | 13-Docosenamide, (Z)- | 3.84 | [M+H]+ | 338.3420 | 338.3423 | 0.86 | 0.8 |
| 15 | PE(16:0/18:2) | 3.93 | [M+H]+ | 716.5248 | 716.5235 | 1.84 | 0.75 |
| 16 | Beta-Sitosterol | 4.03 | [M+H-H2O]+ | 397.3829 | 397.3843 | 1.34 | 0.81 |
| 17 | PC(18:2/18:2) | 4.38 | [M+H]+ | 782.5667 | 782.5700 | 4.19 | 0.94 |
| 18 | Arachidonoyl-Thio-PC | 4.58 | [M+H]+ | 784.5710 | 784.5679 | 3.99 | 0.97 |
| 19 | PC(16:0/18:2) | 4.62 | [M+H]+ | 758.5681 | 758.5700 | 2.48 | 0.91 |
| 20 | PC(16:0/18:1) | 4.74 | [M+H]+ | 760.5866 | 760.5856 | 1.28 | 0.98 |
| 21 | PC(16:0/20:4) | 7.36 | [M+Na]+ | 804.5518 | 804.5519 | 0.16 | 0.75 |

[a]Acceptable range for UPLC-QTOF is 3.0–5.00 ppm. Value > 5.00 ppm requires further verification.

[b]MN/FA: molecular network/ fragmentation analysis.

Small molecules with no cosine score were putatively characterized through molecular networking and fragmentation analysis.

among metabolites and annotation of precursor ions with no library hits. Table 1 includes additional oligosaccharides and dipeptides present in native garlic, annotated using molecular networking and fragmentation analysis (S3 in S3 Fig and S4 in S4 Fig).

Precursor ions m/z 851.26 and 1013.23, with no library hits in GNPS, were connected in a network with putatively annotated oligosaccharides: melibiose, 1-kestose, and stachyose (Fig 1B). These unidentified metabolites were detected in the form of sodiated adducts and showed losses of 162 Da for dehydrated hexose unit, and 180 Da for intact hexose unit (S3 in S3 Fig and S4 in S4 Fig). Therefore, m/z 851.26 and 1013.23 were assigned as sodiated ions of penta- and hexa-oligosaccharides, respectively. These oligosaccharides have been shown to have nutritional benefits and their characterization in Philippine garlic provides information about its dietary contribution.

Simple fructooligosaccharide 1-kestose (glu(α1→2)fru(β1→6)fru) is reported to exhibit bifidogenic activity [24]. It also has a demonstrated positive effect on *Clostridium* clusters IV and XIVa bacteria, which produces butyrate and other short-chain fatty acids essential in the maintenance of gut homeostasis [25]. Stachyose (gal(α1→6)gal(α1→6)glc(α1↔2β)fru), a probiotic, has been recognized to relieve colorectal and hepatic inflammation [26].

Three γ-Glu dipeptides: γ-Glu allyl-cysteine, γ-Glu propenyl-cysteine, and γ-Glu methyl-cysteine, were annotated based on the presence of product ions linked to the cleavage of the amide bond [19]. Common to putatively characterized γ-Glu dipeptides is the m/z 130.05 ion associated with glutamate. γ-Glu propenyl Cys ([M+H]+ = 291.1029, $t_R$ = 1.59 min) was specifically distinguished from its positional isomer, γ-Glu allyl Cys ([M+H]+ = 291.1029, $t_R$ = 1.33

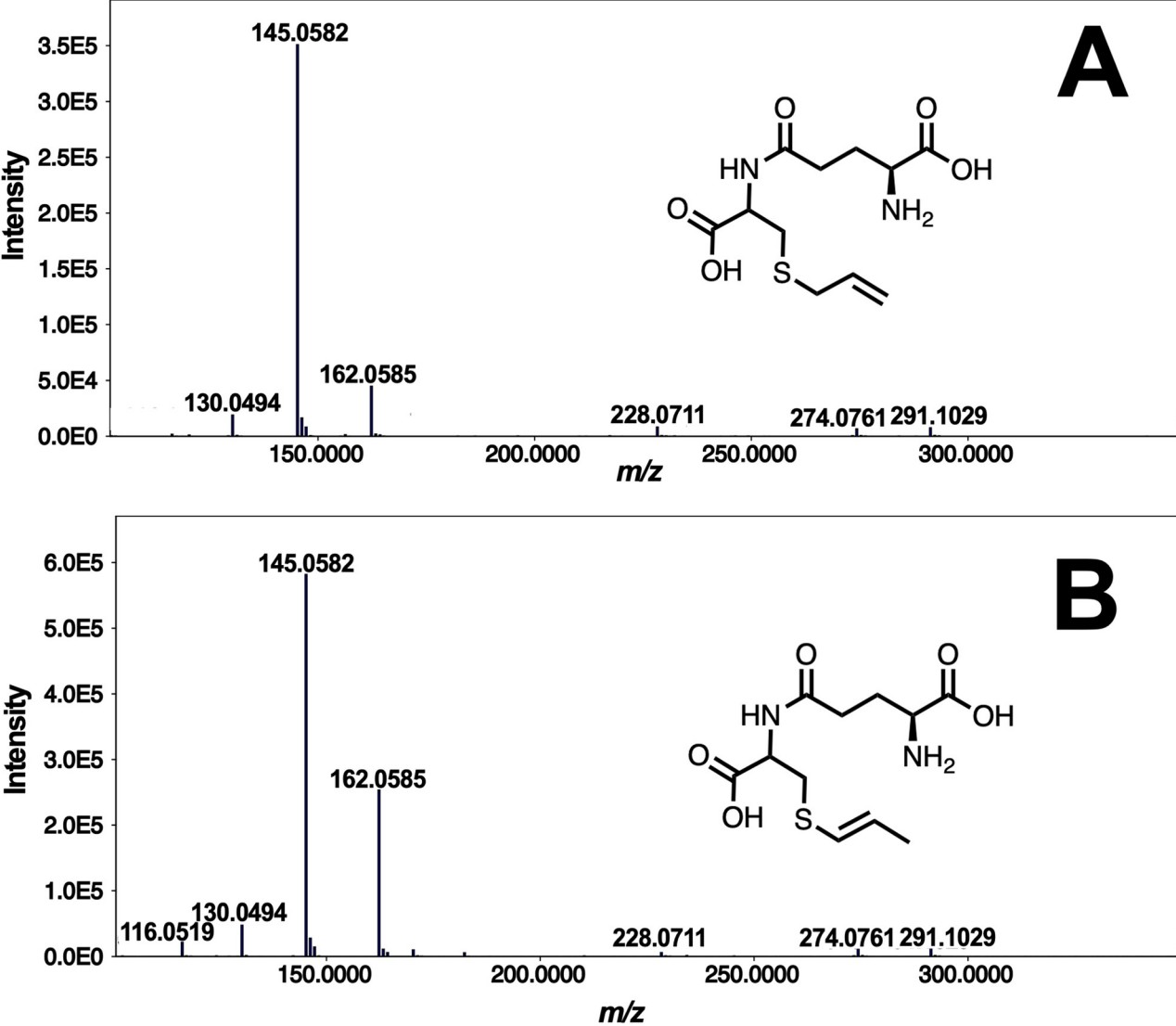

**Fig 2. MS² spectra of isobaric features with m/z 291.10 annotated as γ-glutamyl allyl cysteine and γ-glutamyl propenyl cysteine.** Spectrum A was characterized as γ-glutamyl allyl cysteine while spectrum B is associated with its positional isomer, γ-glutamyl propenyl cysteine ($t_R$ = 1.59 mins).

min) based on the higher intensity of the immonium ion m/z 116.05, and the presence of a unique product ion with m/z 170.08 (Fig 2). A similar observation was described by Nakabayashi et al [23] on a Fourier transform-ion cylcotron resonance analysis of γ-Glu allyl Cys and γ-Glu propenyl Cys. In a separate study [6], fast atom bombardment-MS showed the formation of the pyroglutamate ion at m/z 130 and its complementary ion m/z 162. γ-Glu propenyl Cys also produces a weak dehydration ion [M+H–H2O]+ at m/z 273, together with the parent protonated molecule at m/z 291 [6], which is consistent with the results shown in Fig 2. The proposed fragmentation scheme for distinguishing product ions of the isomeric dipeptides can be found in S3 Fig 2 in S3 Fig.

These γ-Glutamyl dipeptides were initially identified, isolated, and characterized by Ueda et al from water extracts of garlic [5,6] and were the first to propose that these compounds are linked to the *kokumi* flavor [5,6], described by Yang, et. al (2019) as the continuity, thickness,

and mouthful taste of food [27]. These compounds are known to activate oral $Ca^{2+}$ sensing receptors (CaSRs) [28]. γ-Glu methyl cysteine and γ-Glu allyl cysteine putatively characterized in the authenticated samples have been specifically mentioned in previous studies to be flavor-modifying compounds in garlic [29]. Additionally, γ-Glu dipeptides are antioxidants [27] and serve as precursor metabolites to volatile S-alkenyl cysteine sulfoxides such as methiin, propiin, and alliin which contributes to the aroma of garlic [27].

The absence of the bioactive allicin in the samples can be attributed to its high volatility and poor ionization efficiency in both positive and negative ionization modes [29]. Mass spectrometry-based detection of allicin is usually aided by complexation with transition metal ions, such as $Ag^+$ [29] which was not employed in the study. Moreover, allicin has been reported to have a half life of 24 hours in 37˚C [30] implying the possibility of degradation occurring prior to LC-MS analysis as well as probable deactivation of alliinase, the enzyme that catalyzes the oxidation of alliin to allicin. Nevertheless, the allicin precursor, alliin, was putatively annotated in all the samples. Alliin is also known to have antihyperlipidemic and anti-hypertensive benefits [31].

## Multivariate statistical analysis

Some key metabolite features annotated here through molecular networking appeared to be dependent on sampling location (Fig 1B and 1C). Moreover, the relative abundances of metabolites associated with flavor and aroma could account for the difference in taste and smell between samples [32]. To provide an unbiased (i.e. intensity thresholds needed for tandem MS becomes negligible [17] global survey of the garlic, $MS^1$ profiles of all samples were compared using unsupervised multivariate statistics via XCMS Online [18]. Principal component analysis (PCA) represents data in directions of maximal variance [20,33]. This means that the first principal component (PC1; x-axis) has the highest variability followed by the second (PC2; y-axis) and so on [20,33]. Hence, the first and second PCs are usually sufficient in describing the data if these two PCs account for the majority of variance. To correct for the heteroskedastic noise found in MS-generated data, statistical tests used were non-parametric and scores plots from pairwise and multigroup setups were centered and log-transformed [34].

Shang et al., have previously described amino acid, oligosaccharides, dipeptides, and lipids in garlic [31] but this is the first time that these compounds were putatively identified in Philippine cultivars. As selected characteristic features identified in literature, these specific metabolite groups will serve as basis in comparing different cultivars. Moreover, some of these small molecules, such as sulfur-containing dipeptides, have been described in literature for their flavor-modifying effects [3–6].

Authenticated garlic samples BAU and ILAU were compared pairwise to allow the explicit comparisons of their metabolite features, and the results which were used as basis for all successive analyses. ILAU and BAU exhibit a bimodal distribution (Fig 3A), with the scores plot splitting into distinct regions: IB-1 (PC1: -1.0 to -2.0, PC2: 0.0 to -2.5), IB-2 (PC1: -3.5 to -4.5, PC2: -10.0 to -12.5), and IB-3 (PC1: -7.0 to -7.5, PC2: 10.0 to -12.5). The first two regions, IB-1 and IB-2 show clusters with overlaps between ILAU and BAU, implying shared features. Meanwhile, IB-3 is a homogeneous ILAU cluster, indicating that there are features characteristic only to ILAU. In Fig 3B, the biggest contributors to the dispersion of data across PC1 and PC2 are from the three data points that correspond to features m/z 756.6, m/z 780.6, and m/z 784.6. These features were associated with phospholipids. Hence, the difference in levels of lipids within samples from the same location could influence the behavior of the group overall. Lipids provide essential fatty acids and contribute to the food mouthfeel [35]. These compounds also serve as precursors in the Maillard reaction that amplifies flavor and aroma [35].

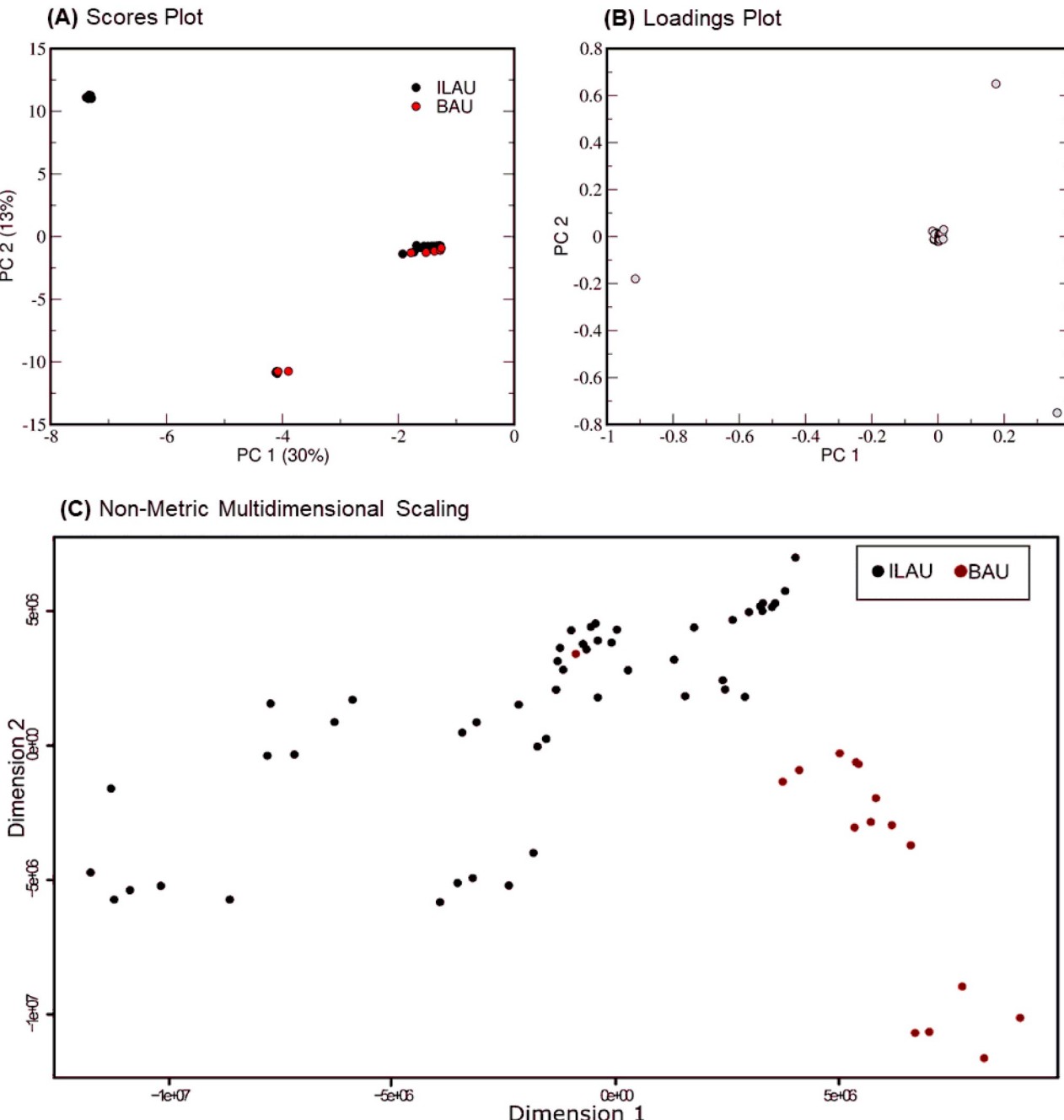

**Fig 3. Comparison between authenticated garlic BAU and ILAU.** Log-transformed PCA scores plot (A) shows that authenticated garlic BAU and ILAU clusters based on shared features.

Consequently, these samples are expected to vary in mouthfeel, aroma, and flavor. Features that have masses smaller than m/z 750 contributed less to the PCs and have clustered at the (0,0) region of the loadings plot and are attributed to dipeptides, amino acids, and oligosaccharides.

Non-Metric Multidimensional Scaling (NMDS) biplot (Fig 3C) provides a simpler visualization of authenticated samples by representing dissimilarities as a function of distance [36].

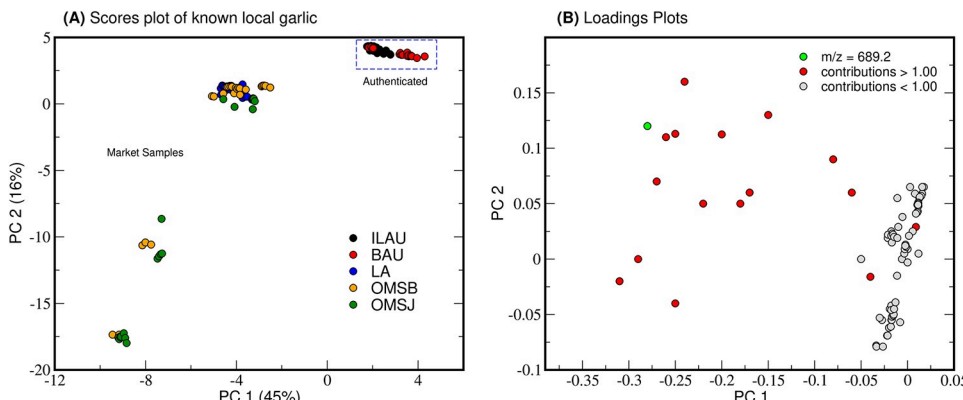

**Fig 4. Multigroup comparison of known native garlic.** Log-transformed PCA scores plot (A) shows that authentic garlic clusters distinctly from other local samples. Mindoro garlic OMSB and OMSJ share features that separate them from the rest of the samples. The variance observed from the samples can be attributed to primarily to metabolites such as oligosaccharide stachyose (m/z 689.2), and other unidentified features (red) as indicated by the loadings plot (B).

The longer distances between ILAU (black) and BAU (red) emphasize their dissimilarities, as expected of samples from different origins. Interestingly, NMDS also shows that dissimilarities also exist between replicates of ILAU as well as in BAU.

By explicitly zooming in at the specific metabolites, it was determined that the clustering of samples is influenced by the levels of sulfur-containing dipeptides. An example of this dipeptide γ-Glu allyl cys (m/z 291.10; p-value: $5.0 \times 10^{-15}$; fold change >1.5), which is present in all local samples, but is upregulated in garlic from Ilocos (ILAU and LA; S5 Figure 6 in S5 Fig & S6 Figure 6 in S6 Fig). Dipeptide γ-Glu methyl cys (m/z 265.10, p-value: $1.99 \times 10^{-12}$, fold change > 3.27) and the metabolite alliin (m/z 178.1, $2.66 \times 10^{-15}$, fold change > 1.68) were also upregulated in ILAU. On the other hand, γ-Glu propenyl cys (p-value: $9.18 \times 10^{-16}$, fold change > 2.24; S5 Figure 5 in S5 Fig) is highly abundant in BAU compared to ILAU. These dipeptides contribute to flavors derived enzymatically in *Allium* species [37]. The presence of these compounds at relatively higher levels in authenticated samples could potentially explain the stronger sensory impression when compared to other cultivars [3,32].

Authenticated cultivars versus local market (Fig 4A) show four distinct regions: R1 (PC1: 1.0 to 4.5, PC2: 2.5 to 5.0), R2 (PC1: -1.0 to -5.0, PC2: 2.0 to -2.0), R3 (PC1: -7.0 to -8.5, PC2: -7.5 to -12.5), and R4 (PC1: -8.5 to -10.0, PC2: -15.0 to -18.0). The ILAU and BAU cluster is represented by R1 while LA, OMSB, and OMSJ gather in R2. R3 and R4 solely describe samples sourced from Mindoro (OMSB and OMSJ). R1 denotes that market samples (R2, R3, and R4) do not share the same characteristics as the authenticated garlic. One of the biggest contributors to the spread of the data is the oligosaccharide stachyose (m/z 689; Fig 4B). Stachyose and 1-kestose were specifically upregulated in LA but downregulated in ILAU and BAU (S5 Figures 2 and 3 in S5 Fig). Oligosaccharides also impart the caramelized notes in food through the Maillard reaction [38]. Furthermore, galacto- and fructooligosaccharides have shown to have a probiotic effect to the human gut microbiota [39] and their characterization in Philippine garlic could provide additional nutritional benefits.

Based on the insights obtained from exploring authenticated and local market samples, a comparison between native and imported garlic (Fig 5A) showed a clear separation across PC1 and PC2. Clustering among native samples is primarily due to higher levels of γ-Glu dipeptides and amino acids, which may be responsible for the enhanced flavor of Philippine garlic [3,32]. Interestingly, some local biological replicates from San Jose (OMSJ) clustered with imported ZAM and IMPSB, which may imply that OMSB outliers have features in common with ZAM

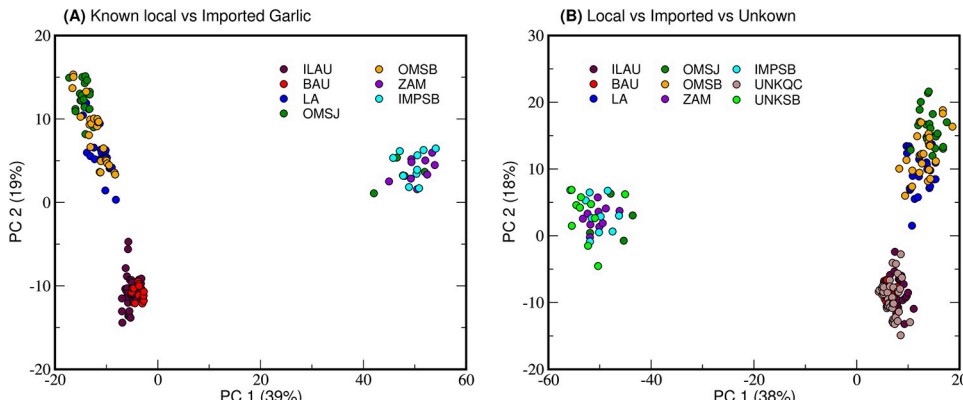

**Fig 5. Scores plot (log, centered) comparing local and imported samples.** Sugars and dipeptides, which influence clustering, are distributed in both local and imported samples.

and IMSB that have a greater loadings weight. Highly abundant metabolites could also swamp out and mask the significant yet less abundant features [34].

From the previous analyses in Figs 3 and 4, identifying the geographical origin of unknown samples becomes possible as native garlic, whether authenticated or market, have distinguishing hallmarks. In Fig 5B, UNKSB and UNKQC were compared to samples of known sources. UNKSB clustered with the imported varieties while UNKQC shared metabolite features with that of the local cultivars. These results signify unsupervised PCA of MS data could be used to determine the origin of samples by comparing their profiles to that of authenticated local and imported garlic.

Through multivariate analysis, it was inferred that the perceived stronger taste of Philippine garlic cultivars, particularly those sourced from Ilocos and Batanes, is potentially due to the higher levels of flavor compounds such as sulfur-containing dipeptides present in native samples compared to the imports. The secondary metabolite profiles of Philippine garlic established in this study provide a basis for its culinary and nutritional value and its molecular signatures can be used to identify geographical origin of garlic.

## Conclusions

Metabolomics and chemometric studies via LC-MS enabled the putative identification of inherent compounds from different Philippine native garlic cultivars as well as in the imported varieties studied here. These are sulfur-containing dipeptides (γ-Glu-Phe, γ-Glu-methyl-cys derivatives), functional oligosaccharides (1-kestose, stachyose, melibiose), lipids and plant hormones (β-sitosterol, indole-3-lactic acid), and volatile molecules (alliin)—all of which contribute to the complex aroma and flavor profile of garlic and have been validated in literature [1,24]. Some of these compounds, particularly amino acids, and sugars, were detected at higher intensities from native garlic cultivars and can be correlated to the enhanced aroma and flavor profile of native garlic. PCA and NMDS highlighted similarities arising from the contributions of shared dipeptides and oligosaccharides between garlic grown in the Philippines. These metabolites influenced the clustering of local garlic and separated them from imported samples. This study is a pioneering work on comparative LC-MS metabolomic and chemometric analyses for a high-value crop cultivated in and imported into the Philippines.

## Supporting information

**S1 Fig.**
(PDF)

**S2 Fig.**
(PDF)

**S3 Fig.**
(PDF)

**S4 Fig.**
(PDF)

**S5 Fig.**
(PDF)

**S6 Fig.**
(PDF)

**S7 Fig.**
(PDF)

**S8 Fig.**
(PDF)

**S9 Fig.**
(PDF)

**S1 Table. Morphometric measurements of native and imported *Allium sativum* cultivars.**
(PDF)

## Acknowledgments

The authors would like to acknowledge the Department of Agriculture Regional Office 02 (DA-RO2)—Batanes Experimental Station through the manager, Mr. Celso Batallones, Mrs. Edelina Rellin (DA-RO2 Philippine Carabao Center) and Mrs. Shirley Abucay for the garlic samples. The authors would also like to thank the Department of Science and Technology-funded Discovery and Development of Health Products Program for the LC-MS facility of the Institute of Chemistry, University of the Philippines, Diliman, and to Mr. Jokent Gaza for his assistance in the computational aspect of this study.

## Author Contributions

**Conceptualization:** Hiyas A. Junio.

**Data curation:** Ralph John Emerson J. Molino, Klidel Fae B. Rellin, Hiyas A. Junio.

**Formal analysis:** Ralph John Emerson J. Molino, Klidel Fae B. Rellin, Ricky B. Nellas, Hiyas A. Junio.

**Supervision:** Ricky B. Nellas, Hiyas A. Junio.

**Validation:** Ralph John Emerson J. Molino, Klidel Fae B. Rellin, Ricky B. Nellas, Hiyas A. Junio.

**Visualization:** Klidel Fae B. Rellin.

**Writing – original draft:** Ralph John Emerson J. Molino, Klidel Fae B. Rellin.

**Writing – review & editing:** Ralph John Emerson J. Molino, Klidel Fae B. Rellin, Ricky B. Nellas, Hiyas A. Junio.

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
