## [Decision Letter · Decision Letter 0]

18 Mar 2021

PONE-D-21-03851

Small in size, big on taste: metabolomics analysis of flavor compounds from Philippine garlic

PLOS ONE

Dear Dr. Junio,

Thank you for submitting your manuscript to PLOS ONE. After careful consideration, we feel that it has merit but does not fully meet PLOS ONE’s publication criteria as it currently stands. Therefore, we invite you to submit a revised version of the manuscript that addresses the points raised during the review process.

We look forward to receiving your revised manuscript.

Kind regards,

Timothy J Garrett, PhD

Academic Editor

PLOS ONE

Journal Requirements:

3. We note that S1 Figure 1 in your submission contains map images which may be copyrighted.

We require you to either (a) present written permission from the copyright holder to publish this figure specifically under the CC BY 4.0 license, or (b) remove the figure from your submission:

a. You may seek permission from the original copyright holder of S1 Figure 1 to publish the content specifically under the CC BY 4.0 license. 

b. If you are unable to obtain permission from the original copyright holder to publish this figure under the CC BY 4.0 license or if the copyright holder’s requirements are incompatible with the CC BY 4.0 license, please either i) remove the figure or ii) supply a replacement figure that complies with the CC BY 4.0 license. Please check copyright information on all replacement figures and update the figure caption with source information. If applicable, please specify in the figure caption text when a figure is similar but not identical to the original image and is therefore for illustrative purposes only.

Additional Editor Comments:

The reviewers have made excellent comments and I hope you take the time to clearly respond to each comment. The differentiation of taste and flavor needs to be very clear. In addition, identification of metabolites without authentic standards means that all metabolites have a very low confidence in identification. Thus, making any biological decisions from those metabolites is very difficult. It is recommend that you follow the metabolomics standards initiative for identification and that key metabolites are identified with standards that can match m/z value, retention time and MS/MS spectra if possible.

Reviewers' comments:

Reviewer's Responses to Questions

**Comments to the Author**

1. Is the manuscript technically sound, and do the data support the conclusions?

Reviewer #1: Partly

Reviewer #2: Partly

2. Has the statistical analysis been performed appropriately and rigorously? 

Reviewer #1: Yes

Reviewer #2: I Don't Know

3. Have the authors made all data underlying the findings in their manuscript fully available?

Reviewer #1: Yes

Reviewer #2: Yes

4. Is the manuscript presented in an intelligible fashion and written in standard English?

Reviewer #1: Yes

Reviewer #2: Yes

5. Review Comments to the Author

Reviewer #1: Pag. 3, line 61 – I suggest author should write the full name of the technique “Liquid chromatography-mass spectrometry” before acronym LC-MS.

Did the author use any sample normalization method for the analyses? Pre-normalization is important for uniform samples across the study. Did the author use, for example, sample mass for pre-normalization? This information needs to be explained at the method section.

Did the author use Quality Control (QC) procedure or internal standards for monitoring the precision of the analytical process involving untargeted metabolomics? This information needs to be explained at the method section.

The author observed the absence of the bioactive compound Allicin in the samples. Could this absence be due a possible inactivation of the allicin-producing enzyme, Alliinase, a naturally occurring enzyme present in the vascular bundle cells of the garlic bulb that it is released when the bulb tissue is crushed?

Reviewer #2: First, I appreciate all the work done in this manuscript.

Not sure how the terms taste and flavor are being used here. Taste should refer to sensations on the tongue (sweet, salty, bitter, sour, umami), while flavor should refer to sensations via retronasal olfaction. Looking through the introduction and references, may of the chemical compounds referred to as influencing taste and flavor have little direct evidence of influencing taste and flavor in garlic through experiments using psychophysics or sensory science. Maybe this point was missed, but it seems many of the compounds are precursors to compounds shown to have taste and flavor functions, like Alliin is a volatile precursor.

It is unclear if 5 bio reps with 5 tech reps per bio rep was used. Please clearly state experimental setup.

Why was a non-parametric analysis chosen?

How were the cultivars authenticated?

I understand qTOF was used but why were no authentic standards used to ensure identification quality?

6. PLOS authors have the option to publish the peer review history of their article (what does this mean?). If published, this will include your full peer review and any attached files.

Reviewer #1: No

Reviewer #2: No

---

## [Author Response · Author response to Decision Letter 0]

29 Apr 2021

From the Editor:

Comment: Please ensure that your manuscript meets PLOS One's style requirements, including those for file naming.

Response: The revised manuscript was edited accordingly. 

Comment: Please review your reference list to ensure that it is complete and correct. If you have cited papers that have been retracted, please include the rationale for doing so in the manuscript text or remove these references and replace them with relevant current references.

Response: The reference list was checked and edited according to the journal’s specifications. 

Comment: We note that S1 Figure 1 in your submission contains map images which may be copyrighted.

Response: We have decided to remove the image to avoid complications with copyright.

Comment: If applicable, we recommend that you deposit your laboratory protocols in protocols.io to enhance the reproducibility of your results. Protocols.io assigns your protocol its own identifier (DOI) so that it can be cited independently in the future.

Response: We have decided not to deposit our protocols on this web-platform.

From Reviewer 1: 

Comment: I suggest the author should write the full name of the technique “Liquid chromatography-mass spectrometry” before the acronym LC-MS. 

Response: This portion of the manuscript was edited as suggested.

Comment: Did the author use any sample normalization method for the analyses? Pre-normalization is important for uniform samples across the study. Did the author use, for example, sample mass for pre-normalization? This information needs to be explained at the method section.

Response: Yes, an important normalization step was performed pre-acquisition in LC-MS. Details with regards to sample normalization method were clarified in the revised manuscript. Please refer to lines 146-147 on page 5 of the revised version.

Comment: Did the author use Quality Control (QC) procedure or internal standards for monitoring the precision of the analytical process involving untargeted metabolomics? This information needs to be explained at the method section.

Response: Yes. We have used quality control samples and internal standards for the analysis and have edited the method section to reflect this (lines 110-112, page 4; lines 135-136, page 5). 

Comment: The author observed the absence of the bioactive compound Allicin in the samples. Could this absence be due a possible inactivation of the allicin-producing enzyme, Alliinase, a naturally occurring enzyme present in the vascular bundle cells of the garlic bulb that is released when the bulb tissue is crushed?

Response: We are not ruling out the possibility of enzymatic action that could inhibit the production of allicin. However, allicin is a volatile compound and its detection is atypical with LC-MS (lines 252-258, page 12).

Comment: Identification of metabolites without authentic standards means that all metabolites have a very low confidence in identification. Thus, making any biological decisions from those metabolites is very difficult. It is recommended that you follow the metabolomics standards initiative for identification and that key metabolites are identified with standards that can match m/z value, retention time and MS/MS spectra if possible.

Response: We acknowledge the limitations of our identification process, however, majority of the compounds we have putatively annotated were categorized as ‘gold’ in the GNPS reference library. These library spectra are derived from purified compounds with NMR and crystallographic data (lines 190-192, page 8). 

Moreover, we have cross-referenced and verified our experimental spectra with those reported in literature that have previously isolated, identified, and characterized the compounds via MS and even NMR (lines 230-236, page 10-11).

From Reviewer 2: 

Comment: Not sure how the terms taste and flavor are being used here. Taste should refer to sensations on the tongue (sweet, salty, bitter, sour, umami), while flavor should refer to sensations via retronasal olfaction. Looking through the introduction and references, many of the chemical compounds referred to as influencing taste and flavor have little direct evidence of influencing taste and flavor in garlic through experiments using psychophysics of sensory science. Maybe this point was missed, but it seems many of the compounds are precursors to compounds shown to have taste and flavor functions, like Alliin is a volatile precursor.

Response: We put great emphasis on gamma-glutamyl dipeptides. Their association with taste is well established through studies that looked at its interaction with nasal/oral targets such as Calcium sensing receptors (CaSRs). These results have also been correlated to sensory evaluation of gamma-glutamyl compounds and their taste-enhancing benefits (Yang, et. al 2019, Toelstede, et al. 2009, Dunket et al., 2007, Yang, et. al, 2021). Based on these studies, review articles describe gamma-glutamyl dipeptides as taste-enhancing compounds that are responsible for the elevated richness and mouthfulness taste of food (Yang, et. al, 2019). However, some studies (Ueda et al., 1990; Nakabayashi et al,. 2016; Amino et al., 2018) which provided data on CaSR assay and sensory evaluation on the compounds isolated from garlic, flavor-modifying was the preferred descriptor.

Comment: It is unclear if 5 bio reps with 5 tech reps per bio rep was used. Please clearly state experimental setup.

Response: The number of biological and technical replicates for each garlic cultivar were clarified in the revised manuscript (lines 99-100, page 3; lines 120-121, page 4).

Comment: Why was a non-parametric analysis chosen?

Response: We have added a clarification in the revised manuscript (lines 151-153, page 7). For untargeted metabolomics data, variances within the population may not be equal. Hence, as powerful as parametric analysis is, it would not be robust enough to handle the inherent heteroscedasticity of LC-MS data.

Comment: How were the cultivars authenticated?

Response: Based on the information provided by the Department of Agriculture Region II, cultivars are authenticated based on seed or clove morphology. The Batanes native cultivar has 12 or more smaller cloves, similar to the Ilocos native variety. Most imported cultivars that have been naturalized in the Philippines have bigger but fewer cloves per bulb (lines 96-98, page 3).

Comment: I understand qTOF was used but why were no authentic standards used to ensure identification quality?

Response: Same answer to the related question of Reviewer 1.

---

## [Editor Report · Decision Letter 1]

5 May 2021

Small in size, big on taste: metabolomics analysis of flavor compounds from Philippine garlic

PONE-D-21-03851R1

Dear Dr. Junio,

We’re pleased to inform you that your manuscript has been judged scientifically suitable for publication and will be formally accepted for publication once it meets all outstanding technical requirements.

Kind regards,

Timothy J Garrett, PhD

Academic Editor

PLOS ONE

Additional Editor Comments (optional):

Thank you for providing detailed responses to each reviewer from the initial submission.  Your responses were very clear and easy to identify.  The review process is very important and I appreciate your careful consideration of the reviews.  
---

## [Editor Report · Acceptance letter]

10 May 2021

PONE-D-21-03851R1 

Small in size, big on taste: metabolomics analysis of flavor compounds from Philippine garlic 

Dear Dr. Junio:

I'm pleased to inform you that your manuscript has been deemed suitable for publication in PLOS ONE. Congratulations! Your manuscript is now with our production department. 

Kind regards, 

on behalf of

Dr. Timothy J Garrett 

Academic Editor

PLOS ONE